# *FBXL7* Body Hypomethylation Is Frequent in Tumors from the Digestive and Respiratory Tracts and Is Associated with Risk-Factor Exposure

**DOI:** 10.3390/ijms23147801

**Published:** 2022-07-15

**Authors:** Diego Camuzi, Luisa Aguirre Buexm, Simone de Queiroz Chaves Lourenço, Rachele Grazziotin, Simone Guaraldi, Priscila Valverde, Davy Rapozo, Jill M. Brooks, Hisham Mehanna, Luis Felipe Ribeiro Pinto, Sheila Coelho Soares-Lima

**Affiliations:** 1Molecular Carcinogenesis Program, Brazilian National Cancer Institute, Rio de Janeiro 20231-050, Brazil; diego.camuzi@dmmd.uzh.ch (D.C.); labuexm@id.uff.br (L.A.B.); rgrazziotin@inca.gov.br (R.G.); sguaraldi@inca.gov.br (S.G.); lfrpinto@inca.gov.br (L.F.R.P.); 2Department of Pathology, Dental School, Fluminense Federal University, Niterói 24040-110, Brazil; silourenco@id.uff.br; 3Division of Radiation Oncology, National Cancer Institute, Rio de Janeiro 20230-240, Brazil; 4Endoscopy Section, National Cancer Institute, Rio de Janeiro 20230-130, Brazil; 5Pathology Division, National Cancer Institute, Rio de Janeiro 20230-130, Brazil; pvalverde@inca.gov.br (P.V.); davyrapozo@gmail.com (D.R.); 6Institute of Head and Neck Studies and Education (InHANSE), Institute of Cancer and Genomic Sciences, University of Birmingham, Birmingham B15 2TT, UK; j.m.brooks@bham.ac.uk (J.M.B.); h.mehanna@bham.ac.uk (H.M.); 7Biochemistry Department, Biology Institute, State University of Rio de Janeiro, Rio de Janeiro 20511-010, Brazil

**Keywords:** DNA methylation, cancer, squamous cell carcinoma, esophagus, head and neck, FBXL7, AURKA, Survivin, HPV

## Abstract

Squamous cell carcinoma is the main histological tumor type in the upper aerodigestive tract (UADT), including the esophagus (ESCC) and the head and neck sites, as well as the oral cavity (OCSCC), larynx (LSCC) and oropharynx (OPSCC). These tumors are induced by alcohol and tobacco exposure, with the exception of a subgroup of OPSCC linked to human papillomavirus (HPV) infection. Few genes are frequently mutated in UADT tumors, pointing to other molecular mechanisms being involved during carcinogenesis. The F-box and leucine-rich repeat protein 7 (FBXL7) is a potential tumor-suppressing gene, one that is frequently hypermethylated in pancreatic cancer and where the encoded protein promotes the degradation of AURKA, BIRC5 and c-SRC. Thus, the aim of this study was to evaluate the methylation and expression profile of *FBXL7* in the UADT and the gene’s association with the clinical, etiological and pathological characteristics of patients, as well as the expression of its degradation targets. Here we show that the *FBXL7* gene’s body is hypomethylated in the UADT, independently of histology, but not in virus-associated tumors. *FBXL7* body methylation and gene expression levels were correlated in the ESCC, LSCC, OCSCC and OPSCC. Immunohistochemistry analysis showed that FBXL7 protein levels are not correlated with the levels of its degradation targets, AURKA and BIRC5, in the UADT. The high discriminatory potential of *FBXL7* body hypomethylation between non-tumor and tumor tissues makes it a promising biomarker.

## 1. Introduction

The SCF (SKP1-Cul1-F-box) complex is composed of a basic structure of S-phase kinase-associated protein 1 (SKP1), the E3 ligase, RBX1 (also known as ROC1), and cullin 1, as well as variable F-box proteins that confer substrate selectivity by targeting a distinct subset of substrates for ubiquitylation [1]. F-box proteins have pivotal roles in multiple cellular processes through the ubiquitylation and subsequent degradation of target proteins [2]. The dysregulation of F-box protein-mediated proteolysis leads to human malignancies [2]. The F-box and leucine-rich repeat protein 7 (*FBXL7*) is a potential tumor suppressor gene (TSG), which encodes an E3 ligase protein that promotes the ubiquitination and consequent proteasome degradation of target proteins. Although few studies to date have focused on the FBXL7, the Aurora kinase A (AURKA) [3], c-SRC [4] and BIRC5/Survivin [5] proteins have been identified as its degradation targets. FBXL7 targets are widely studied oncoproteins [6,7,8] that are linked to increased proliferation, metastasis and resistance to apoptosis, and their overexpression is recurrently reported in solid tumors, leading to a worse prognosis [9,10,11,12].

In cancer, TSG can be silenced by epigenetic mechanisms such as aberrant DNA methylation [13]. This epigenetic mechanism can regulate gene expression, depending on its levels and the affected gene region [14]. In tumors, for example, TSG silencing has been associated with promoter hypermethylation and gene-body hypomethylation [15]. Due to its role in driving TSG inactivation, widespread and local aberrant DNA methylation profiles have been reported and represent an important field in oncology for generating biomarkers and new therapeutic options [16,17].

Tumors from the upper aerodigestive tract (UAT) show few recurrently mutated genes, suggesting that other molecular mechanisms take part in their development. In these tumors, DNA methylation is intricately linked with risk-factor exposure [18] and malignant transformation [19]. The esophagus (ESCC) and, among the head and neck organs, the oral cavity (OCSCC), larynx (LSCC) and oropharynx (OPSCC) are most commonly affected by squamous cell carcinomas [20,21]. Besides their shared histology and origin from the same epithelial lining, for most of these cancers, alcohol and tobacco represent the main risk factors [21]. The exception is a sub-group of OPSCC that is linked to human papillomavirus (HPV) infection [22], which shows a different methylation profile and therapeutic response relative to other head and neck squamous cell carcinomas (HNSCC) [23,24].

The overexpression of AURKA and BIRC5 has already been reported in HNSCC [9,12] and ESCC [11,25], being linked to malignancy [11,26,27], resistance to treatment [11,28] and worse prognoses [9,11,12,25]. Despite the dysregulation of AURKA and BIRC5 in these tumors, to the best of our knowledge, no study has yet focused on molecular alterations to the gene encoding their degradation regulator, FBXL7. Recently, *FBXL7* promoter hypermethylation was associated with its downregulation, c-SRC induction and tumor progression in prostate and pancreatic cancers [4]. Based on this finding, the aim of this study was to evaluate the methylation profile and regulation of *FBXL7* expression in HNSCC and ESCC, and their association with the patients’ clinical, etiological, and pathological characteristics. We also evaluated whether FBXL7 protein levels were altered by and correlated with AURKA and BIRC5.

## 2. Results

### 2.1. FBXL7 Methylation Levels in Esophageal and Head and Neck Squamous Cell Carcinomas

The data used in this subsection were generated by the authors. Initially, we evaluated the methylation profile of all methylome probes within *FBXL7* (Appendix A shows the characteristics of patients, while Appendix A shows the location of each probe in *FBXL7*), and observed a common gene body hypomethylation in ESCC, LSCC, OCSCC and OPSCC, relative to their respective non-tumor surrounding tissues (NTST) or tissues from donors without cancer (OPSCC, exclusively). In every anatomical site, this methylation profile could differentiate between tumor and non-tumor tissues after unsupervised clustering. On the other hand, clusterization does not seem to be influenced by alcohol/tobacco consumption history, the tumor stage or patient age. The higher lethality of ESCC compared to any other head and neck tumor [29], as well as the high prevalence of ESCC as a second primary tumor in HNSCC patients [30], led us to select the methylome probe for validation by pyrosequencing, based on its discriminative values of tumor vs. NTST in patients with ESCC. According to the ESCC microarray analysis, the cg11339964 probe, mapped to the gene body, had the highest accuracy (Appendix A). *FBXL7* body hypomethylation was confirmed by pyrosequencing in ESCC (*p* < 0.0001), LSCC (*p* < 0.0001) and OCSCC (*p* = 0.0004), relative to NTST (Figure 1B), when the genomic position referring to the probe cg11339964 was assessed. The pyrosequencing and microarray results showed a high correlation (rho = 0.9767, *p* < 0.0001) (Figure 1C).

As we observed a predominance of gene body hypomethylation in tumor samples, we sought to verify how much it would represent the tumor mass, in view of the intratumoral heterogeneity using ESCC as a model. *FBXL7* body hypomethylation was almost universal in the tumor regions relative to the NTST of the same patient (Figure 1D). Only one NTST sample from one patient (Pt2) showed *FBXL7* methylation levels that were similar to tumors. In view of the apparent homogeneous hypomethylation of *FBXL7*, we verified its discriminatory potential, regarding the CpG relative to the cg11339964 probe, between the NTST and tumor by applying a receiver-operating characteristics (ROC) curve analysis, showing accuracies of 88.1%, 85.2% and 94.4% for ESCC, LSCC and OCSCC, respectively (Figure 1E).

Next, the associations between *FBXL7* methylation levels and the sociodemographic and clinical characteristics of ESCC, LSCC, OCSCC and OPSCC The Cancer Genome Atlas (TCGA) patients were verified. In OCSCC patients, *FBXL7* hypomethylation was associated with an advanced tumor stage (*p* = 0.00923). OPSCC from never-smokers showed higher methylation levels when compared to ever-smokers (*p* = 0.039). *FBXL7* hypomethylation was associated with worse survival in OPSCC (*p* = 0.0194); however, significance was lost after multivariable Cox regression adjustment by age, tumor stage and HPV status (*p* = 0.2242, HR: 0.45 (0.12–1.64)). For all other analyses in all tumor types, no significant associations were found (Appendix A).

### 2.2. FBXL7 Methylation in Tumors from Different Anatomic Sites and Its Association with Etiology

ESCC and HNSCC originate from the same epithelial lining and share a common etiology related to alcohol and tobacco [31]. At the same time, DNA methylation is deeply linked with tissue differentiation [32] and environmental exposure [33,34]. Therefore, to evaluate whether *FBXL7* hypomethylation is associated with tissue-specific malignant transformation and/or risk factor exposure, we verified its methylation profile in the same genomic position (cg11339964) in tumors from the head and neck, digestive tract and lungs of subjects with different histologies and associated with different risk factors. Data used in this subsection were retrieved from publicly available databases, as specified in Materials and Methods.

Esophageal adenocarcinoma (EAC) cases showed *FBXL7* hypomethylation relative to the non-tumor esophageal mucosa (*p* < 0.0001), Barrett’s esophagus (*p* = 0.0319) and esophageal mucosa from individuals with gastroesophageal reflux disease (GERD, *p* < 0.0001) (Figure 2A). Barrett’s esophagus cases also displayed lower *FBXL7* methylation levels compared with non-tumor tissues (*p* = 0.0055). *FBXL7* methylation levels did not vary according to EAC patients’ alcohol or smoking histories, which are secondary risk factors for this tumor [35]. Lung adenocarcinomas (LUAD) also showed *FBXL7* hypomethylation (*p* = 0.0494) and ever-smokers presented lower methylation levels relative to never-smokers (*p* = 0.0003) (Figure 2B). We also observed *FBXL7* hypomethylation in lung squamous cell carcinoma (LUSCC) compared to NTST (*p* < 0.0001); however, no differences regarding smoking habits were detected (Figure 2C). Colorectal cancer was also included in this analysis, showing *FBXL7* hypomethylation relative to NTST (*p* < 0.0001) (Figure 2D).

Since *FBXL7* hypomethylation was detected in tumors from different histologies and an association with risk-factor exposure was observed in some cases, we sought to evaluate whether this profile could also be seen in viral infection-associated HNSCC. In nasopharyngeal squamous cell carcinoma (NPSCC), associated with Epstein-Barr Virus (EBV) infection, the dataset GSE62336 showed no methylation differences between tumor and NTST tissues (*p* = 0.2389) (Figure 2E). The human papillomavirus-positive (HPV+) OPSCC showed higher *FBXL7* methylation levels when compared to HPV-negative (HPV-) OPSCC (commonly associated with alcohol and tobacco history), in TCGA-HNSC (*p* < 0.0001) and Brazilian patients (*p* < 0.0001) (Figure 2F). In order to further explore the impact of HPV infection on *FBXL7* methylation, we evaluated cervical squamous cell carcinoma samples from GSE99511 and found no differences between normal tissues, cervical squamous intraepithelial neoplasia grade 3 (CIN3) tissues and carcinomas (Figure 2G).

### 2.3. FBXL7 Gene Expression Is Correlated with Its Gene Body Methylation Levels

Using the RNASeq data from TCGA patients (publicly available dataset), we performed a correlation analysis between all available Infinium 450K (Illumina, San Diego, CA, USA) probes mapped to *FBXL7* and gene expression in ESCC, LSCC, OCSCC and OPSCC patients from the TCGA cohort (Figure 3). In general, *FBXL7* promoter methylation was not significantly associated with gene expression, except for two probes in OPSCC and one probe in LSCC showing inverse correlations. The methylation levels of gene body probes were more commonly significantly correlated with gene expression, with all significant correlations being direct in all tumors. For cg11339964, a significant correlation with *FBXL7* expression was observed in ESCC and LSCC.

We looked for possible associations between the clinical and sociodemographic characteristics of TCGA patients and *FBXL7* expression levels in ESCC, LSCC, OCSCC and OPSCC (Appendix A). In OPSCC, never-smokers showed lower *FBXL7* mRNA levels compared to smokers (*p* = 0.011). In the case of the other tumors, we did not find any statistically significant association.

### 2.4. The FBXL7 Protein Is Overexpressed and Is Not Related to HPV Status or Its Target Proteins in UADT Tumors

Finally, to better understand the relationship between FBXL7 and its targets, AURKA and BIRC5, in UADT tumors, we analyzed their protein levels in Brazilian patients diagnosed with ESCC, LSCC and OPSCC (Figure 4A–C, data generated by the authors). The proportion of cells with FBXL7 cytoplasmic positivity was higher in ESCC and OPSCC, relative to their respective NTST. When nuclear staining was considered, the percentage of positive cells was lower in ESCC and LSCC when compared to NTST. AURKA cytoplasmic positivity was higher in ESCC, LSCC and OPSCC, while for BIRC5, the same profile was observed in ESCC and OPSCC. The nuclear staining of AURKA was not detected in LSCC NTST, but the proportion of positive cells was higher in ESCC and OPSCC, relative to their NTST. For BIRC5, no significant differences in nuclear positivity were observed for any of the tumors analyzed. Figure 4C shows a summary of all findings. We did not observe any difference in FBXL7 protein staining according to HPV status, both in the cytoplasm and in the nuclei of OPSCC cells (Figure 4D).

Finally, we correlated the percentage of positive cells for the three proteins; only BIRC5 and AURKA showed a significant correlation with OPSCC (rho = 0.34, *p* = 0.0053, Figure 4E). The association between cytoplasmatic protein levels and patients’ clinical and sociodemographic characteristics (Appendix A) showed higher FBXL7 cytoplasmatic positivity in the early stages of LSCC, higher BIRC5 in LSCC patients who did not report alcohol consumption, better prognoses in ESCC patients with lower BIRC5 positivity, and better prognosis in OPSCC patients with higher AURKA levels.

## 3. Discussion

*FBXL7* gene functions are poorly understood, but the gene’s role as a tumor suppressor has been proposed based on its target proteins, AURKA, BIRC5 and c-SRC. These are oncoproteins that are overexpressed in different tumor types and their targeting by FBXL7 suggests that the latter may indirectly control the cell cycle [3] and apoptosis [5]. Although the overexpression of FBXL7 targets has been reported in ESCC [11,25] and HNSCC [9,12], to date, no data are available on FBXL7 expression. Based on this finding, the present study aimed at investigating *FBXL7* dysregulation and the molecular regulatory mechanisms that are potentially involved in ESCC and HNSCC. Recently, *FBXL7* promoter hypermethylation was associated with the FBXL7 silencing and overexpression of its target protein, c-SRC, increasing metastasis promotion in prostate and pancreatic cancers [4]. However, the methylation profile of *FBXL7* in ESCC and HNSCC is unknown, and the impacts of its body methylation levels are not understood. Here we showed a consistent *FBXL7* gene body hypomethylation in ESCC, LSCC, OCSCC and OPSCC, significantly correlated with mRNA expression. Our results further showed that the same aberrant DNA methylation profile is observed in tumors from the aerodigestive tract from other histologies and is associated with different risk factors, but not in neoplasias where the development is driven by viral infections. Therefore, *FBXL7* gene body hypomethylation seems to be induced by specific exposures and might be a useful biomarker of specific carcinogenesis mechanisms.

DNA methylation is the most widely studied epigenetic mechanism, and it is closely linked with tissue transformation and risk factor exposure [34,36]. ESCC and HNSCC have an aberrant DNA methylation profile [37,38]; we show here that the *FBXL7* gene body is hypomethylated in both tumor types, among other tumors of the digestive and respiratory tracts. This hypomethylation seems to be consistent throughout the esophageal tumor mass, as shown in our intratumoral heterogeneity analysis. However, as a limitation of the present study, a similar assessment was not possible in the other tumor types, due to organ-specific characteristics. According to the direct correlation between gene body methylation and gene expression that is observed in all tumor types, decreased *FBXL7* mRNA levels are likely to be detected in ESCC, LSCC, OCSCC and OPSCC, relative to their NTST. Gene body hypomethylation was already associated with the activation of intragenic enhancers, spurious transcription, or repetitive elements [39,40,41], which may influence the rate of polymerase elongation and alternative splicing [42,43]; this should be explored in future studies.

Although the consequences of *FBXL7* body hypomethylation are not clear, it is surprising that only those tumors not associated with viral infection showed this profile. Previous studies have shown that HPV-negative HNSCC presents a global hypomethylation profile compared with HPV-positive HNSCC [44,45]. In addition, HPV oncoproteins (E6 and E7) were shown to promote the activity of DNA-methyltransferases (DNMTs). E6 induces DNMT1 expression via p53 repression in cervical cells [46], while HNSCC cells treated with a DNA demethylation agent show a decrease in HPV gene expression with induced p53-dependent apoptosis [47]. E7 has the same effect on DNMT1 via pRb repression [48,49,50]; additionally, it directly binds to DNMT1, enhancing its activity [49,50]. This might explain, at least in part, the differences observed in HPV+ tumors, but the detection of a similar profile in EBV-associated tumors raises the question of whether this impact on DNA methylation is driven by any viral infection. Viral oncoproteins are usually considered to be the drivers of viral carcinogenesis, but epigenetic alterations may also play a role in this equation.

FBXL7 protein positivity was generally lower in the nuclei of tumor cells, as expected by the gene body hypomethylation and the direct correlation with gene expression but was higher in the cytoplasm. This observation raises the question of whether gene body methylation could regulate the expression of different *FBXL7* expression isoforms, coding different proteins with distinct cellular localization. Indeed, three *FBXL7* isoforms were already described (Appendix A), but their association with protein expression and cellular compartments has not been explored. Additionally, we cannot rule out the impact of post-translational modifications and protein-protein interactions on FBXL7 levels. For example, AURKA can contribute to BIRC5 stability by directly targeting FBXL7 [51]. Despite the positive correlation between AURKA and BIRC5 that was observed in OPSCC in our sample set, FBXL7 did not show a correlation with AURKA. Furthermore, the results at the protein level suggest that in the evaluated tumors, FBXL7 is not the main pathway for AURKA and BIRC5 degradation. In HNSCC cell lines, AURKA can be phosphorylated on serine 51 and stabilized [52]. In addition, it is speculated that FBXL7 only degrades limited amounts of AURKA in the centrosomes at specific steps during the cell cycle [53]. BIRC5 can be stabilized by HSP90, which is overexpressed in UADT tumors [54,55].

The discriminatory accuracy between tumors and non-tumor tissues, together with the homogeneity of *FBXL7* gene body hypomethylation within the tumor mass, make it a potential diagnostic biomarker. Unfortunately, most ESCC and HNSCC tumors are diagnosed at a late stage, so our case series included few early-stage tumors, raising the question of when this hypomethylation would occur. Nevertheless, the observation of lower methylation levels in some NTST samples and in premalignant lesions (Barrett’s esophagus), together with the lack of association with patients’ survival, suggest that this is an early change and is not associated with tumor progression. Finally, although the *FBXL7* gene body hypomethylation was not associated with the sociodemographic and clinical characteristics evaluated, it could be used as a specific biomarker for the diagnostic and etiological classification of tumors from the digestive and respiratory tracts. For example, OPSCC caused by an HPV infection show a better treatment response and overall survival rate than those cases caused by alcohol and tobacco [56,57]; therefore, their stratification can help in guiding the treatment path [57].

## 4. Materials and Methods

### 4.1. Patients

All patients included in the study had a histopathological diagnosis of squamous cell carcinoma, confirmed by a pathologist, and none of them received chemotherapy or radiotherapy before sample collection.

Snap-frozen biopsies of tumors and non-tumor surrounding tissues (NTST) from ESCC patients (*n* = 70) were obtained from two Brazilian hospitals, Hospital Universitário Pedro Ernesto of the Universidade do Estado do Rio de Janeiro (HUPE-UERJ, Rio de Janeiro, Brazil) and the Instituto Nacional de Câncer (INCA, Rio de Janeiro, Brazil). Snap-frozen samples from laryngeal squamous cell carcinoma (LSCC, *n* = 56) and oral cavity squamous cell carcinoma (OCSCC, *n* = 16) patients were also collected at INCA. OPSCC patients from INCA (*n* = 78) and the PET-Neck trial (England, *n* = 8) were included in the study, from whom snap-frozen or formalin-fixed and paraffin-embedded (FFPE) samples were retrieved.

From four ESCC patients, biopsies were collected to analyze intratumoral heterogeneity. Two biopsies, one superficial (1) and another deep (biopsy-on-biopsy scheme) (2) were collected from each third of the tumor (Distal/Tumor A, Middle/Tumor B and Proximal/Tumor C) and two biopsies were collected from the adjacent non-tumor tissue, one in reference to the proximal part and another distal to the tumor. Thus, for each patient included in the assessment of intratumoral heterogeneity, six tumor biopsies and two from the adjacent non-tumor tissue were collected where possible. Patient 4 had a very large tumor blocking the esophageal lumen, which did not allow the complete passage of the endoscope during the exam; therefore, the collection of distal non-tumor surrounding tissues and tumor tissues was not possible. Thus, the tumor biopsies were collected from the proximal and middle parts of the tumor.

This research project was approved by the Ethics Committees of all institutions involved. All experiments were performed according to the Helsinki Declaration.

### 4.2. Methylome Analysis

Methylome analysis was performed as previously described [19]. Briefly, DNA was extracted from ESCC and HNSCC tumors and their respective non-tumor adjacent tissues (NTST), all snap-frozen, with the DNeasy Blood & Tissue Kit (Qiagen, Hilden, Germany), as follows: 16 NTST and 24 ESCC; 12 NTST and 20 LSCC; seven NTST and 15 OCSCC; and 8 OPSCC. In addition, methylomes were also performed using DNA extracted from nine tonsils, taken from individuals without cancer (patients undergoing tonsillectomies as a sleep dyspnea treatment) and eight OPSCC samples, all FFPE, with the QIAamp DNA FFPE Tissue Kit (Qiagen). After extraction, the DNA was treated with sodium bisulfite using the EZ DNA Methylation kit (Zymo Research, Irvine, CA, USA) and the methylation profile was evaluated by the microarray platform, Illumina HumanMethylation450 BeadChip (Illumina, San Diego, CA, USA), following the manufacturer’s specifications.

After checking the internal controls with GenomeStudio Software (Illumina, San Diego, CA, USA), analysis of the .idats files was performed by methylumi [58] in the R software suite (v3.6) to obtain the average beta values. Low-quality (*p* > 0.05), cross-reactive and polymorphic probes [59] were removed. Adjustments for color (lumi [60]) and probe biases were performed using the BMIQ method (watermelon [61]) and, if necessary, a correction for batch effect was applied. The idats files from the Cancer Genome Atlas patients (TCGA-ESCA and TCGA-HNSC projects) were obtained using the TCGAbiolinks package (v2.16.1) and were processed using the ChAMP package (v2.18.2). The previously described filters were applied to obtain the average beta values. Clinical and pathological data from the TCGA patients were obtained from the cBioPortal (https://www.cbioportal.org/, accessed on 16 May 2022) with the cgdsr package (v1.3.0).

### 4.3. Immunohistochemical Staining

Three 3-μm serial histological sections were prepared for each paraffin block. When available in the same slide, both the tumor and NTST were evaluated, as follows: 21 ESCC and 22 NTST; 47 LSCC and 34 NTST; and 78 OPSCC and 51 NTST. Immunohistochemistry was performed with FBXL7 (sc-374319, Santa Cruz, Dallas, Texas, USA), AURKA (ab1287, abcam, Cambridge, UK) and BIRC5 (sc-17779, Santa Cruz) antibodies. Antigen retrieval was performed with Trilogy™ for 50 min at 98 °C. The detection was performed using Novolink™ Max Polymer Detection System (Leica, Newcastle, UK), following the manufacturer’s protocol. The evaluation was performed by an expert pathologist; slides with staining in less than 5% of the tumor cells were not included in the statistical analyses. Digital images were generated using the Aperio ScanScope CS Slide Scanner (Aperio Technologies, Leica, Newcastle, UK).

### 4.4. DNA Methylation Analysis by Pyrosequencing

A total of 500 ng of genomic DNA was extracted from NTST and respective tumors from ESCC and HNSCC patients, as follows: 42 ESCC patients, 42 LSCC patients, 11 OCSCC patients and 82 OPSCC patients. After extraction, the genomic DNA was treated with sodium bisulfite using the EZ DNA Methylation-Gold Kit (Zymo Research, Irvine, CA, USA). Hot-start PCR was performed with Platinum^®^ Taq DNA Polymerase High Fidelity (Invitrogen, Waltham, Massachusetts, USA) and two different pairs of primers were used, one for snap-frozen tissues (FBXL7_Fresh assay) and the second for FFPE tissues (FBXL7_FFPE assay): FBXL7_Fresh_F - 5’ GAGATAGATGTATATTTGGTTG 3’ (GRCh37/hg19, chr5:15,645,438-15,645,459) and FBXL7_Fresh_R - 5’ biotin-ATCCTAATAACATATACCCAC 3’ (GRCh37/hg19, chr5:15,645,679-15,645,699), FBXL7_FFPE_F - 5′ GTTTTTTGATTGATATTTTTTGTTTT 3′ (GRCh37/hg19, chr5:15,645,616-15,645,641) and FBXL7_FFPE_R - 5′ biotin-ATCCTAATAACATATACCCAC 3′ (GRCh37/hg19, chr5:15,645,679-15,645,699). The thermal cycling program consisted of an initial denaturation for 15 min at 95 °C, followed by 50 cycles of 30 s at 95 °C, 30 s at 52 °C for the FBXL7_Fresh assay and 50 °C for the FBXL7_FFPE assay, then 30 s at 72 °C and a final extension for 10 min at 72 °C. Pyrosequencing was carried out using a PyroMark Q96 ID, in accordance with the manufacturer’s protocol (Qiagen, Hilden, Germany), with the following sequencing primer: FBXL7_sequencing - 5’ ATAAATGTATTTTTT 3’. The target CpGs were evaluated by converting the resulting pyrograms into numerical values for peak heights. In both assays, the CpG corresponding to the genomic location of cg11339964 Infinium Human Methylation 450K BeadChip (Illumina, San Diego, CA, USA) probe was evaluated.

### 4.5. Human Papillomavirus (HPV) Status

The HPV status of OPSCC samples was assessed, as previously described [62]. Briefly, tumor genomic DNA was used for amplifying the HPV16 E6 gene by quantitative PCR (qPCR); FFPE OPSCC samples were used to evaluate p16 positivity via immunohistochemistry. Only those samples positive for both E6 amplification and p16 labeling were considered HPV-positive, while samples with positivity in only one of the tests or negative in both assays were considered HPV-negative. The TCGA patient’s HPV status was obtained at http://firebrowse.org/(accessed on 10 August 2020).

### 4.6. mRNA Expression Analysis

The RNA-seq data from TCGA patients were downloaded using the TCGAbiolinks package (v2.16.1). Pre-processed FPKM values (level-3 data) from the TCGA-ESCA and TCGA-HNSC projects were used.

### 4.7. Gene Expression Onminibus Data Analysis

The Gene Expression Omnibus (GEO) database was assessed to retrieve the DNA methylation data (Illumina HumanMethylation450 BeadChip methylation microarray platform) from the OPSCC (GSE38271), esophageal adenocarcinoma (GSE72874), lung adenocarcinoma (GSE56044), lung squamous cell carcinoma (GSE68825), cervical carcinoma (GSE99511), nasopharyngeal carcinoma (GSE62336) and colorectal adenocarcinoma (GSE101764) datasets.

### 4.8. Statistical Analysis

Statistical tests were performed according to data distribution, assessed via the Kolmogorov–Smirnov normality test. Data with parametric distribution were compared by ANOVA and paired or unpaired *t*-tests. Data with nonparametric distribution were assessed via the Mann–Whitney, Wilcoxon or Kruskal–Wallis tests. Survival analyses were performed in R software (v3.6) with the Survival package (v3.2-3). The Kaplan–Meier survival curves were generated for the univariate survival analysis, and the statistical significance between the two groups was calculated using the log-rank test. When necessary, multivariate Cox regression was applied. For all correlations, we adopted the Spearman test. Heatmaps were designed with *ComplexHeatmap* v2.0.0. Differences with a *p*-value < 0.05 were considered statistically significant.

## 5. Conclusions

*FBXL7* gene body hypomethylation is common in cancers from the upper aerodigestive tract, independently of histology, with virus-associated tumors being the only exceptions. *FBXL7* body methylation was positively correlated with gene expression in all evaluated tumors and lower nuclear protein staining was observed in ESCC and LSCC, relative to the non-tumor surrounding tissue. A higher percentage of AURKA nuclear positivity was observed in the same samples. The high discriminatory potential of *FBXL7* methylation levels between non-tumor and tumor tissues, together with the intratumor homogeneity of this molecular alteration, make it a promising biomarker.

## Figures and Tables

**Figure 1 ijms-23-07801-f001:**
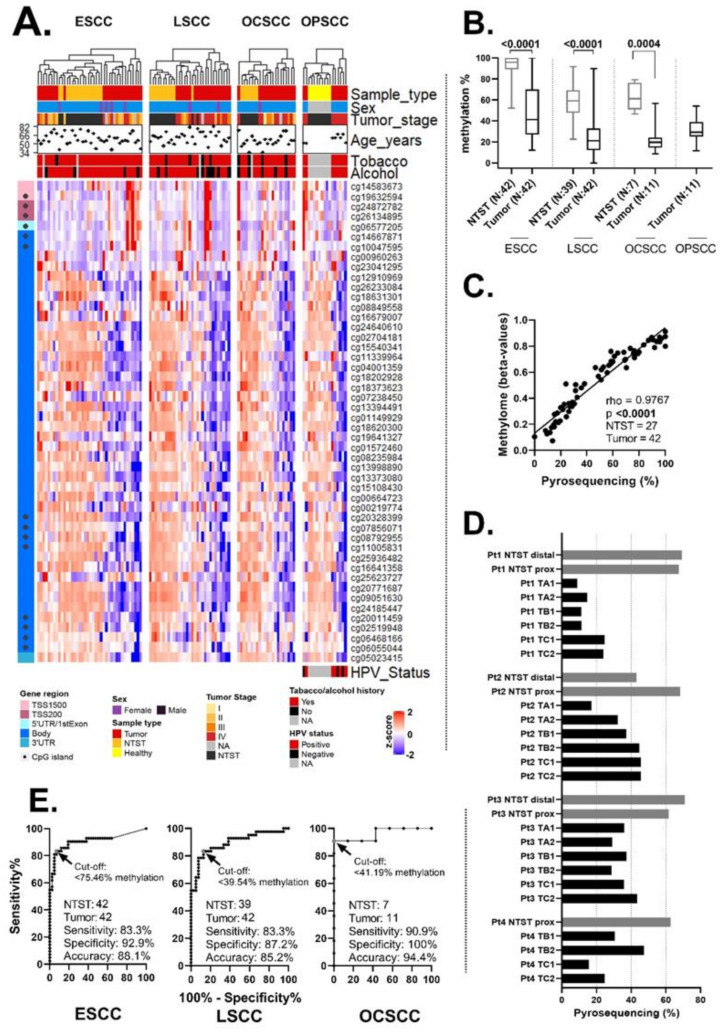
The *FBXL7* methylation profile in esophageal (ESCC), larynx (LSCC), oral cavity (OCSCC) and oropharynx squamous cell carcinoma (OPSCC). (**A**) Heatmap showing the methylation profile of all methylome probes located within *FBXL7,* with unsupervised clustering by samples (vertical). Horizontally, the probes are organized according to gene region. (**B**) Boxplots showing the validation by pyrosequencing of the methylome findings in the genomic position referring to the cg11339964 probe in ESCC, LSCC, OCSCC and OPSCC patients. (**C**) The correlation between *FBXL7* methylation levels assessed by pyrosequencing (methylation percentage) and methylome (beta-values). (**D**) Bar plots showing the intratumor heterogeneity of *FBXL7* methylation levels in ESCC. Four patients were included in this analysis, and non-tumor surrounding tissue samples were collected from the distal (NTST distal) and proximal (NTST prox) parts of the esophagus, while tumor samples were obtained superficially (1) or profoundly (2) from the distal (TA), middle (TB) and proximal (TC) parts of the tumor. (**E**) Receiver-operating characteristics (ROC) curves, showing the discriminative power of *FBXL7* methylation levels (cg11339967 probe methylation levels assessed by pyrosequencing) between non-tumor surrounding tissue and tumor tissue in ESCC, LSCC and OCSCC. Note: NTST, non-tumor surrounding tissue; NA, not available. TSS1500 and TSS200 refer to up to 1500 bp and 200 bp upstream of the transcription start site, respectively.

**Figure 2 ijms-23-07801-f002:**
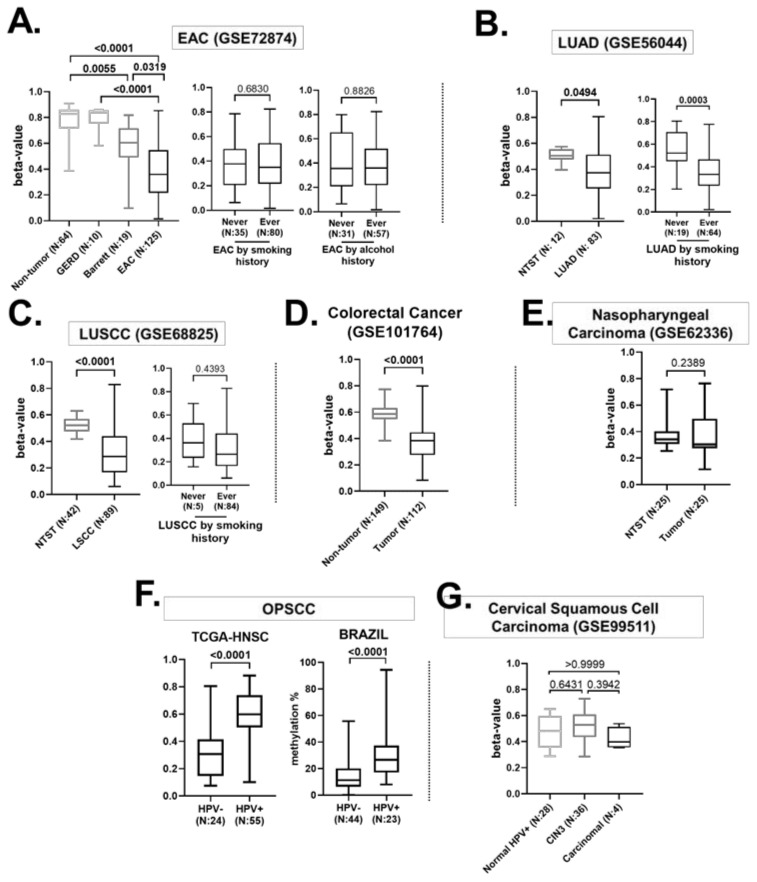
*FBXL7* methylation (cg11339964) in aerodigestive tract tumors from different histologies and that are associated with different risk factors. (**A**) On the left is a boxplot showing *FBXL7* methylation levels in non-tumor esophageal mucosa from healthy individuals and from individuals with gastroesophageal reflux disease (GERD), metaplastic Barrett’s esophagus (Barrett) and esophageal adenocarcinoma (EAD). The EAD samples were further stratified by smoking history (boxplot in the center) and alcohol history (boxplot on the right). Data were retrieved from GSE72874. (**B**) Boxplots showing *FBXL7* methylation levels in lung adenocarcinoma (LUAD) and non-tumor surrounding tissue (NTST) and in LUAD samples, stratified by smoking history (data retrieved from GSE56044). (**C**) Boxplots showing *FBXL7* methylation levels in lung squamous cell carcinoma (LUSCC) and non-tumor surrounding tissue (NTST), and in LUSCC samples stratified by smoking history (data retrieved from GSE68825). (**D**) Boxplot showing *FBXL7* methylation levels in colorectal adenocarcinoma samples and non-tumor tissue (data retrieved from GSE101767). (**E**) Boxplots showing *FBXL7* methylation levels in nasopharyngeal squamous cell carcinoma and non-tumor surrounding tissue (NTST) from GSE62336. (**F**) Boxplots showing *FBXL7* methylation levels in oropharynx squamous cell carcinoma (OPSCC), stratified by HPV status, from the TCGA dataset (analyzed by microarray) and Brazil (present study, analyzed by pyrosequencing). (**G**) Boxplots showing *FBXL7* methylation levels in cervical squamous cell carcinoma, cervical squamous intraepithelial neoplasia grade 3 (CIN3), and normal HPV infection (GSE99511).

**Figure 3 ijms-23-07801-f003:**
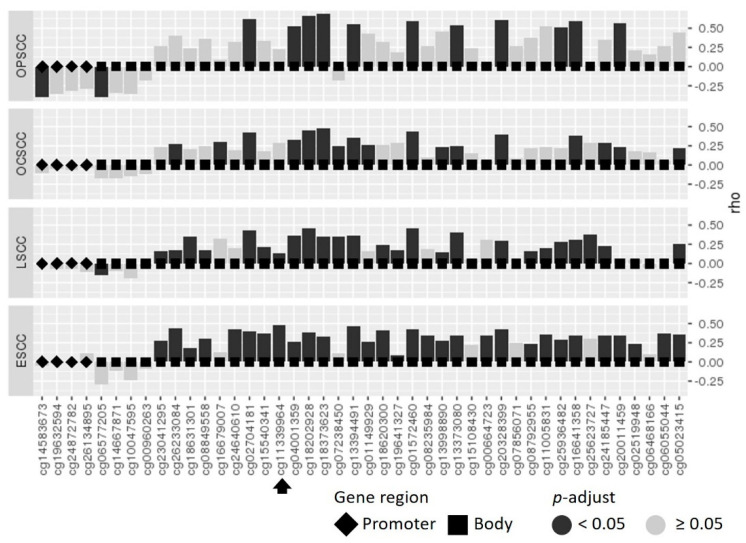
Gene body methylation is correlated with *FBXL7* expression in UADT tumors. Correlation analysis between *FBXL7* methylation (all Illumina HumanMethylation450 BeadChip (Illumina, San Diego, CA, USA) probes available) and mRNA expression levels in esophageal (ESCC), larynx (LSCC), oral cavity (OCSCC) and oropharynx squamous cell carcinoma (OPSCC) from TCGA datasets. The black arrow indicates the probe that is validated by pyrosequencing in the present study.

**Figure 4 ijms-23-07801-f004:**
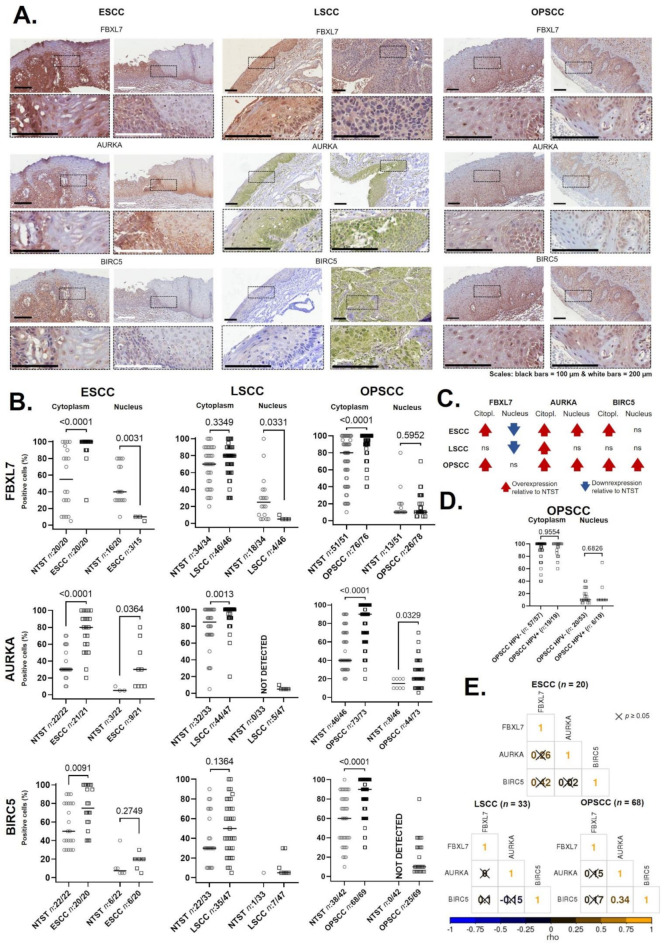
Evaluation of protein levels of FBXL7 and its degradation targets AURKA and BIRC5 in upper aerodigestive tract tumors by immunohistochemistry. (**A**) Representative images of immunohistochemical staining of FBXL7, AURKA and BIR5 in esophageal (ESCC), larynx (LSCC) and oropharynx squamous cell carcinoma (OPSCC). (**B**) Dot plots showing the percentage of positive cells for each marker in the cytoplasm or nucleus of all tumor types analyzed and their respective non-tumor surrounding tissues. The *n*-values represent the number of patients’ slides considered in this analysis from the total of slides stained. Slides with positivity in less than 5% of the cells were not included (see Section 4). (**C**) Summary of the findings in the protein analyses. Red arrows represent higher positivity in the tumor relative to NTST; blue arrows represent lower positivity in the tumor relative to NTST; ns, no significant differences. (**D**) The proportion of FBXL7 positive cells in OPSCC, according to HPV status. (**E**) Correlation between positive cytoplasmic staining of FBXL7, AURKA and BIRC5 in ESCC, LSCC and OPSCC.

## Data Availability

All data generated and used here are available in public databases: The Cancer Genome Atlas (TCGA-ESCA and TCGA-HNSC projects) and Gene Expression Omnibus (GSE178212 for ESCC; GSE178216 for OSCC, GSE178218 for LSCC; and GSE178219 for OPSCC).

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
