# Peer review of "FBXL7 Body Hypomethylation Is Frequent in Tumors from the Digestive and Respiratory Tracts and Is Associated with Risk-Factor Exposure"

_ijms, 2022, doi:10.3390/ijms23147801_

Round 1

Reviewer 1 Report

The research in the manuscript "FBXL7 body hypomethylation is frequent in tumors from the  digestive and respiratory tracts and associated with risk factor exposure" is based on two approaches: a) analyses of experimental data obtained on the bench, b) analyses of data deposited in several publicly available data bases. The authors should very clearly state the source of data in at the beginning of every subsection of the Results section.

The introductory part does not present sufficient information with respect to SCFs (SKP1-Cul1-F-box), which play a role in phosphorylation-dependent ubiquitination of target proteins.

With respect to 2.1.:

There should be a Table showing the distribution of patients with respect to all parameters presented in Figure 1A (anatomic location, sex, age, smoking, alcohol abuse, HPV presence). I am aware that some of these data are presented in Suppl. Table 3. Still, the Table should be functionally connected with Figure 1A, which itself should be presented in a far higher resolution and in a far more logical fashion.

Gene body methylation was explored through the methylation status of the probe cg11339964. Please make the position of the probe clear according to TV1; NM_012304. Does  Figure 1E relate to ROC values obtained with cg11339964? That is not clear.  Are TSS500 and TSS200  the gene regions upstream of  TSS (+1)? That is not clear. Please, make clear the topographic position of the primers listed in  subsection 4.4. with respect to NM_012304 and cg11339964, or chromosome 5, GRCh38.p14 Primary Assembly (NC_000005.10).

With respect to tumor heterogeneity, the authors used  espohageal cancer samples which can be relatively easily separated from the non/tumorous part of the tissue. They should have tried to perform the same analyses using  tumors excised from a narrow anatomical location ( e.g .larynx).

With respect to 2.2.:

While I appreciate the amount of data offered, I see no need for presenting all of them, because most of them were not further discussed.

With respect to 2.3.:

Please provide the genomic position of probes, so the reader can check whether the spots with aberrant methylation reside close to the end/beginning of exons. This should be considered from the point of aberrant splicing and presence of numerous FBXL7 splice variants. Some of them are presented on Suppl. Figure 2 which itself is not sufficiently self-explanatory. The textual listing of data does not contribute to an understanding of the basic process. It just repeats what was already shown.

With respect to 2.4.:

The resolution of IHC images is very low and must be improved. I am not sure why the authors did not apply a immunohistochemical reactivity score, which is based on the percentage of positive cells and the intensity of the signal.

This material must be significantly improved. Some claims are very superficial ("Immunohistochemistry analysis suggest that FBXL7 is not inducing the degradation of AURKA and BIRC5 in UADT"). It is not focused and does not go deeply into the mechanicistic aspect of the study. It is surprising that the authors performed methylation analyses but they did not quantify the FBXL7 transcripts in the same samples. That approach would be closer to the postulates of personalized medicine.

Thank you.

Author Response

The research in the manuscript "FBXL7 body hypomethylation is frequent in tumors from the  digestive and respiratory tracts and associated with risk factor exposure" is based on two approaches: a) analyses of experimental data obtained on the bench, b) analyses of data deposited in several publicly available data bases. The authors should very clearly state the source of data in at the beginning of every subsection of the Results section.

Answer: We thank the reviewer for all the comments and suggestions. They surely improved the manuscript and made it more clear to the reader. Regarding the source of data, statements at the beggining of each subsection were added.

The introductory part does not present sufficient information with respect to SCFs (SKP1-Cul1-F-box), which play a role in phosphorylation-dependent ubiquitination of target proteins.

Answer: We thank the reviewer for making this observation. More information about the SCF complex is now provided in the introduction.

With respect to 2.1.:

There should be a Table showing the distribution of patients with respect to all parameters presented in Figure 1A (anatomic location, sex, age, smoking, alcohol abuse, HPV presence). I am aware that some of these data are presented in Suppl. Table 3. Still, the Table should be functionally connected with Figure 1A, which itself should be presented in a far higher resolution and in a far more logical fashion.

Answer: We added a new Supplementary File 1 providing a summary of all characteristics of interest for each subgroup of patients included in the methylome analysis, according to tumor type (Figure 1A). Regarding Figure 1, it was restructured and its resolution was improved. Indeed, the Figures’ resolution withing the text was not very high, but we also uploaded the Figures as individual files.

Gene body methylation was explored through the methylation status of the probe cg11339964. Please make the position of the probe clear according to TV1; NM_012304. Does  Figure 1E relate to ROC values obtained with cg11339964? That is not clear.  Are TSS500 and TSS200  the gene regions upstream of  TSS (+1)? That is not clear. Please, make clear the topographic position of the primers listed in  subsection 4.4. with respect to NM_012304 and cg11339964, or chromosome 5, GRCh38.p14 Primary Assembly (NC_000005.10).

Answer: ROC curves in Figure 1E were calculated for each tumor type using cg1133964 probe methylation levels, evaluated by pyrosequencing, of samples presented in Figure1B. We have added this information to the main text and figure legend. Regarding TSS1500 and TSS200 probes, they are up to 1,500 bp and 200 bp upstream the transcription start site, respectively. This information was made clearer in Figures' legends. Primers' positions were also added to the main text.

With respect to tumor heterogeneity, the authors used  espohageal cancer samples which can be relatively easily separated from the non/tumorous part of the tissue. They should have tried to perform the same analyses using  tumors excised from a narrow anatomical location ( e.g .larynx).

Answer: We agree with the reviewer that it would have been interesting to perform the intratumor heterogeneity analyses with other tumor types in addition to esophageal cancer. However, due to the the much more narrow and the smaller size of the other organs analyzed, this was not possible in our settings. We added this point as a limitation of our study in the discussion section.

With respect to 2.2.:

While I appreciate the amount of data offered, I see no need for presenting all of them, because most of them were not further discussed.

Answer: We believe section 2.2 is an important part of our study. After identifying FBXL7 hypomethylation in esophageal and head and neck tumors, we began to investigate its possible causes. Since DNA methylation is associated with both cell identity and risk factor exposure, characteristics shared between the tumors initially evaluated in our study, we assessed other tumors with similar characteristics, but also tumors that have a different histology and/or etiology, as follows. This analysis led us to conclude that chemical carcinogenesis is likely to be the cause of FBXL7 hypomethylation and we have a paragraph in the Discussion section bringing hypotheses on the mechanisms by which viral infections might induce different DNA methylation profiles.

With respect to 2.3.:

Please provide the genomic position of probes, so the reader can check whether the spots with aberrant methylation reside close to the end/beginning of exons. This should be considered from the point of aberrant splicing and presence of numerous FBXL7 splice variants. Some of them are presented on Suppl. Figure 2 which itself is not sufficiently self-explanatory. The textual listing of data does not contribute to an understanding of the basic process. It just repeats what was already shown.

Answer: We thank the reviewer for raising this very important point. In figure 1A, we now provide further information regarding probes' positions, showing the corresponding gene positions on the left side of the heatmap. Additionally, we have added information to Supplementary Figure 2 (now named Supplementary File 2) regarding the location of the probes within the gene and its isoforms. Illumina also provides detailed information about each probe in its Microarray manifest files (https://support.illumina.com/downloads/infinium_humanmethylation450_product_files.html).

With respect to 2.4.:

The resolution of IHC images is very low and must be improved. I am not sure why the authors did not apply a immunohistochemical reactivity score, which is based on the percentage of positive cells and the intensity of the signal.

Answer: We agree that the Figures’ resolution withing the manuscript file was not very high, but we also uploaded the Figures as individual files. Regarding the IHC analysis, we chose to use only the percentage of positive cells because when positivity was observed, the intensity of the signal was generally high. Also, since the experiments were performed in different batches (due to the high number of samples and markers), this type of analysis helped us to avoid possible batch-effects.

This material must be significantly improved. Some claims are very superficial ("Immunohistochemistry analysis suggest that FBXL7 is not inducing the degradation of AURKA and BIRC5 in UADT"). It is not focused and does not go deeply into the mechanicistic aspect of the study. It is surprising that the authors performed methylation analyses but they did not quantify the FBXL7 transcripts in the same samples. That approach would be closer to the postulates of personalized medicine.

Answer: We thank the reviewer for these comments. We revised our statements and changed some of them in order to make them more faithful to our findings. For example, the statement mentioned by the reviewer was replaced by "Immunohistochemistry analysis shows that FBXL7 protein levels are not correlated with the levels of its degradation targets AURKA and BIRC5 in UADT". We also agree that the analysis of FBXL7 transcripts in the same samples we evaluated DNA methylation levels would reinforce our findings. Unfortunately, this was not possible because we used biopsies as the source of samples, which restricted the number of molecular analyses to be performed. In order to try to overcome this limitation, we used publicly available data on FBXL7 mRNA and DNA methylation levels obtained from the same samples. We believe this was important for both validating our results in an independent set of samples and assessing the correlation between gene expression and methylation.

Reviewer 2 Report

The manuscript “FBXL7 body hypomethylation is frequent in tumors from the digestive and respiratory tracts and associated with risk factor exposure” by Camuzi et al., demonstrates interesting and novel results.

The Authors were studying the altered methylation and expression levels of the FBXL7 gene in UADT (the upper aerodigestive tract) tumors, compared to non-tumor surrounding tissues (NTST) or tissues from donors without cancer (used as controls).

They have explored the methylation signature of the FBXL7 gene in the mentioned tissue samples, to verify the methylation mark (one CpG: cg11339964) within the FBXL7 gene body as a potential candidate for diagnostic and/or prognostic biomarker of UADT. The Authors demonstrated that the methylation level of the cg11339964 within the FBXL7 gene body differs between tumor and “non-tumor” tissues, suggesting its usefulness as a potential candidate for a diagnostic biomarker of UADT.

The paper is well-written. The methods employed, obtained results, and data analysis are reasonable. The methylation profile of the selected tumor and “non-tumor” samples was evaluated by the Illumina 450K. The results obtained from this microarray platform should be validated using e.g. pyrosequencing - the Authors validated using pyrosequencing only one CpG corresponding to the genomic location of cg11339964 from Illumina 450K. Why only this CpG located within the FBXL7 gene body has been chosen and tested/validated by the Authors? Therefore, the Authors should be more careful with the conclusions in the manuscript, including the term “FBXL7 body methylation” or “FBXL7 body hypomethylation, as they validated the methylation level only for one CpG within the gene body. Moreover, there is nothing more mentioned about the location of the tested CpG (cg11339964), i.e. that it is located within an enhancer region (it should be discussed). Please provide an explanation on the CpG chosen for validation. Consider showing a gene map for the gene fragment including this CpG.

Minor comments:

- Page 4, line 106: should it be Pt2 instead of Pt3 (according to Figure 1d, “Pt2 NTST distal”)?

- Why in Figure 1D, for patient 4 (Pt4) are there the results for only one NTST sample, and only distal and proximal parts of tumor shown, less than for Patients 1-3 (Pt1-Pt3)?

- In Figure 1B (ESCC), the boxplot for NTST samples (n=42) shows the range of methylation (Q1-Q3) around 90-100%, and in Figure 1D (ESCC) for NTST samples (Pt1-Pt4), the range of methylation is around 42-72% - why?

- In Figure 1B, why is the number of NTST samples not equal to LSCC and OCSCC samples?

- Page 9, lines 260-263: the sentence ”E6 induces DNMT1 expression by p53 repression in cervical cells [42], and HNSCC cells treated with DNA demethylation agent have a decrease of HPV genes expression with induced p53 apoptosis [43].” should be corrected – please consider the following alterations: “decrease in…”  and “induced p53-dependent apoptosis”.

Author Response

The manuscript “FBXL7 body hypomethylation is frequent in tumors from the digestive and respiratory tracts and associated with risk factor exposure” by Camuzi et al., demonstrates interesting and novel results.

The Authors were studying the altered methylation and expression levels of the FBXL7 gene in UADT (the upper aerodigestive tract) tumors, compared to non-tumor surrounding tissues (NTST) or tissues from donors without cancer (used as controls).

They have explored the methylation signature of the FBXL7 gene in the mentioned tissue samples, to verify the methylation mark (one CpG: cg11339964) within the FBXL7 gene body as a potential candidate for diagnostic and/or prognostic biomarker of UADT. The Authors demonstrated that the methylation level of the cg11339964 within the FBXL7 gene body differs between tumor and “non-tumor” tissues, suggesting its usefulness as a potential candidate for a diagnostic biomarker of UADT.

The paper is well-written. The methods employed, obtained results, and data analysis are reasonable. The methylation profile of the selected tumor and “non-tumor” samples was evaluated by the Illumina 450K. The results obtained from this microarray platform should be validated using e.g. pyrosequencing - the Authors validated using pyrosequencing only one CpG corresponding to the genomic location of cg11339964 from Illumina 450K. Why only this CpG located within the FBXL7 gene body has been chosen and tested/validated by the Authors? Therefore, the Authors should be more careful with the conclusions in the manuscript, including the term “FBXL7 body methylation” or “FBXL7 body hypomethylation, as they validated the methylation level only for one CpG within the gene body. Moreover, there is nothing more mentioned about the location of the tested CpG (cg11339964), i.e. that it is located within an enhancer region (it should be discussed). Please provide an explanation on the CpG chosen for validation. Consider showing a gene map for the gene fragment including this CpG.

Answer: We would like to thank the reviewer for all valuable comments and suggestions. They surely helped improving the manuscript. The validation of the cg11339964 probe methylation levels by pyrosequencing was performed because this probe showed the highest accuracy among all microarray probes to distinguish ESCC and non-tumor surrounding tissues. The higher lethality of ESCC compared to any other head and neck tumor, as well as the high prevalence of ESCC as a second primary tumor in head and neck patients, led us to select this methylome probe for validation by pyrosequencing. This information was added to the manuscript and as a Supplementary File 1. We also added the location of the probe within FBXL7 gene in the Supplementary File 2.

Minor comments:

- Page 4, line 106: should it be Pt2 instead of Pt3 (according to Figure 1d, “Pt2 NTST distal”)?

Answer: We thank the reviewer for bringing to our attention this important typo. This has been corrected in the manuscript.

- Why in Figure 1D, for patient 4 (Pt4) are there the results for only one NTST sample, and only distal and proximal parts of tumor shown, less than for Patients 1-3 (Pt1-Pt3)?

Answer: Patient 4 had a very large tumor blocking the esophageal lumen, which did not allow the complete passage of the endoscope during the exam and, therefore, the collection of distal non-tumor surrounding and tumor tissues was not possible. Thus, the tumor biopsies were collected from the proximal and middle parts of the tumor, and this was corrected in the manuscript. We thank the reviewer for raising this point, which made us notice these inconsistencies. These details were added to the manuscript.

- In Figure 1B (ESCC), the boxplot for NTST samples (n=42) shows the range of methylation (Q1-Q3) around 90-100%, and in Figure 1D (ESCC) for NTST samples (Pt1-Pt4), the range of methylation is around 42-72% - why?

Answer: In Figure 1B we show the variation of DNA methylation across tumor samples from different patients, while Figure 1D illustrates the variation across samples from the same patient. Methylation levels from samples shown in Figure 1D were not plotted in Figure 1B. Therefore, the difference of patients and number of samples included in each analysis resulted in a different methylation range.

- In Figure 1B, why is the number of NTST samples not equal to LSCC and OCSCC samples?

Answer: For head and neck cancer, depending on the tumor size, it might be a challenge to collect non-tumor surrounding tissue. In the case of the larynx, non-tumor surrounding samples are usually collected from the opposite wall relative to the tumor. This is not possible for oral cavity, especially when tumors are located at the tongue (most commonly affected subsite).  Therefore, for some patients the collection of non-tumor surrounding tissue was not possible, what explains the different numbers observed between tumor sites.

- Page 9, lines 260-263: the sentence ”E6 induces DNMT1 expression by p53 repression in cervical cells [42], and HNSCC cells treated with DNA demethylation agent have a decrease of HPV genes expression with induced p53 apoptosis [43].” should be corrected – please consider the following alterations: “decrease in…”  and “induced p53-dependent apoptosis”.

Answer: The sentence was corrected in the revised version of the manuscript.

Round 2

Reviewer 1 Report

I thank authors for the effort they take to improve the manuscript.

Reviewer 2 Report

The Authors have addressed all of my comments and concerns in the revised version. I have no additional comments.